# A Network Maturity Mapping Tool for Demand-Driven Supply Chain Management: A Case for the Public Healthcare Sector

**Munyaradzi Bvuchete** [1], **Sara Saartjie Grobbelaar** [2,*] **and Joubert van Eeden** [1]

1 Department of Industrial Engineering, Stellenbosch University, Stellenbosch 7600, South Africa; bvuchete@gmail.com (M.B.); jveeden@sun.ac.za (J.v.E.)
2 DST-NRF Centre of Excellence in Scientometrics and Science, Technology and Innovation Policy (SciSTIP), Stellenbosch University, Stellenbosch 7600, South Africa
* Correspondence: ssgrobbelaar@sun.ac.za

**Abstract:** The healthcare supply chain is a complex adaptive ecosystem that facilitates the delivery of health products to the end patient in a cost-effective way. However, low forecast accuracy and high demand volatility in healthcare supply chains have resulted in an increase in stockouts, operational inefficiencies, poor health outcomes, and a significant increase in supply chain costs. To cope with these challenges, organisations are trying to adopt demand-driven supply chain management (DDSCM) operating practices which have been established in other sectors such as the telecommunications, fruit, and flower industries. However, previous studies have not considered these practices in the healthcare industry, and hence no methodologies exist that support the implementation of these practices in this context. Moreover, current studies present cases where the focus has been on improving and expanding individual organisational performance, but no supply chain network-level studies exist on the healthcare industry. Therefore, this paper provides a network-level analysis when addressing DDSCM in the healthcare industry. A grounded theory-based approach coupled with a conceptual framework analysis process was used to leverage a systematized literature review methodology with the development of a network maturity mapping tool for DDSCM which was validated in the public healthcare sector.

**Keywords:** public healthcare supply chains; demand-driven supply chain management; maturity models; segmentation; product classification

## 1. Introduction

A large proportion of total healthcare expenditure in low- and middle-income countries (LMIC) is spent on medicines [1]. However, due to ineffective procurement and supply chain systems in the public healthcare sector (PHS), the supply of medicines to health facilities is often interrupted, resulting in frequent stockouts [2]. In the case of antiretroviral (ARVs) and tuberculosis drugs in particular, stockouts increase the potential risk of virus reactivation and the risk of emergence of drug-resistant strains [3]. There thus exists a need to develop agile and efficient healthcare supply chain systems [4] to ensure efficient health service delivery. In supply chain systems, low forecast accuracy and high demand variability force organisations to increase safety stock levels or move products from one location to another on an expedited basis. However, these initiatives hurt operational efficiency and increase supply chain costs. To cope with this scenario, many organisations are trying to move from a pure supply push strategy, which is only driven by forecast, to a demand-driven supply chain management (DDSCM) strategy, which is driven by actual customer demand [5]. DDSCM is viewed as consisting of management practices that coordinate the supply chain, starting with the end customer and working backwards towards the manufacturer [6]. Moreover, it is a pull-based approach consisting of coordinated technologies and processes that collect, analyse, and share real-time demand and inventory information across all supply chain partners [7]. This strategy ensures a better

balance between supply and customer demand, thereby delivering expected customer service levels and supply chain efficiency [8], as well as ensuring the availability of the right medicines when and where they are most needed [2].

Although DDSCM has been established in sectors such as the computer, fruit, flower, telecommunications, transport, beef, and fashion industries [9–11] to manage supply chain complexity, demand volatility, and uncertainty [12], previous works have not comprehensively considered this strategy in public healthcare supply chain networks (SCNs) in developing countries. Healthcare supply chains need to move towards an integrated "demand pull" strategy that enables manufacturers to have visibility regarding the actual consumption of medicines [5]. Hence, the focus that is required is to ensure that supply chains in the PHS are engineered to match demand requirements [13]. Only when the requirements and constraints of the customer are understood can an attempt to develop a strategy that will meet the needs of both the supply chain and the end customer be made [11]. Moreover, previous studies on DDSCM strategy have only taken the viewpoint of an individual supply chain node, while network-level studies on the public healthcare sector are still scarce [14]. Specifically, these studies have focused on improving and expanding the individual supply chain performance evaluation of organisations' suppliers, distributors, and customers, but no network maturity mapping tools (NMMTs) exist to help manage typical complexities in the management of supply chain networks (SCNs) [15]. Consequently, tackling supply chain complexities such as low forecast accuracy and high demand variability in public healthcare SCNs to date has been beyond the existing tools and techniques. SCNs are characterized by nonlinear interactions and strong interdependencies between supply chain nodes [16].

A network perspective improves and facilitates the understanding of the complexities of interconnected supply chain nodes and feedback loops within public healthcare SCNs. By adopting this DDSCM orientation, supply chain nodes may attain benefits, which include enhanced responsiveness, improved delivery performance, and reduced inventory and supply chain costs [17]. Therefore, the research question for this paper is as follows:

(i) How can a healthcare supply chain network deal with low forecast accuracy and high demand volatility using DDSCM practices that have been applied in other industries?
(ii) What are the components that make up DDSCM?
(iii) What does a network maturity mapping tool for DDSCM look like?
(iv) Is the network maturity mapping tool for DDSCM applicable in the healthcare context?

The main aim of this paper is to design a network maturity mapping tool for DDSCM that is applicable in the healthcare context to support supply chain optimization efforts. The structure of the paper is as follows. Section 2 elaborates on the theoretical background of the DDSCM concept and presents a literature review on maturity models as well as supply chain mapping (SCM). Section 3 describes the methodology for the development and application of the DDSCM NMMT. Section 4 outlines the results and discussion. Section 5 discusses the gaps and contributions of the paper. The management implications of the study are outlined in Section 6. Lastly, concluding remarks with indications for future research are outlined in Section 7.

## 2. Theoretical Background and Literature Review

### 2.1. Healthcare Supply Chain Network

The pharmaceutical component represents a significant part of healthcare costs. For example, in the United States, health care costs were up to $3.2 trillion in 2015, while healthcare supply chain inefficiency spending represented 48% of the total annual cost. Despite the size and importance of this industry around the world, the areas of healthcare, logistics, and supply chain management (SCM) have been given relatively little attention. Thus, among those processes with potential for improvement, healthcare supply network design is important due to considerable resources and costs, as well as their effect on the service level of the healthcare systems [18].

South Africa is still burdened by a high human immunodeficiency virus (HIV) prevalence (13.06% in 2018) and a high incidence of cases of active TB (567 cases of active tubercle bacillus (TB) per 100,000 people in 2017). This means that stockouts of medicines will have severe impacts on these affected populations. Medicine stockouts thus increase the risk of emergence of drug-resistant strains in the health delivery system and also result in high levels of morbidity and mortality [19]. The key role of healthcare supply chains in mitigating the morbidity and mortality of a disaster has already been recognized by practitioners [20]. In the real-world situation, supply chains operate in an uncertain environment due to external supply, supply deliveries, and customer demand. Hence, the reliability of the healthcare supply chain should be taken into consideration along with the cost objectives [18].

The need to develop agile and efficient healthcare supply chain systems is thus important [19]. Ideally, healthcare supply chain design should address disruptive events by upgrading the reliability of its constituents [18]. Current management systems and practices in public healthcare supply chains are less and less able to cope with the growing complexities of low forecast accuracy and demand variability. This results in medicine stockouts, poor healthcare outcomes, high supply chain costs, and operational inefficiencies [17]. In another case, reports on Hurricane Andrew in 1992 state that about 1500 of the patients who were referred to a homestead field hospital encountered some limitations regarding pharmaceutical supplies because all sources of antibiotics, insulin, and tetanus toxoid had been depleted in the first day after the disaster. Akbarpour [20] proposes a min–max robust model to table demand uncertainty. Their numerical results reveal that using the min–max robust model enhances the pharmaceutical relief network's effectiveness and efficiency considerably. The proactive pre-positioning of supplies of medicine as well as the planned distribution of medicine items in an efficient and effective way will reduce the morbidity and mortality of disasters [20].

A healthcare supply chain is an ecosystem consisting of a combination of organisations, people, technology, activities, information, and resources that facilitate the delivery of healthcare products, vaccines, and other medicines from the manufacturer to the end-patient in a cost-effective way. The prime purpose of public healthcare supply chains is to ensure agile healthcare delivery systems for the citizens of a country, regardless of their geographic location and in response to diversified patient needs [19]. A strong healthcare delivery system cannot function without a well-designed, well-operated, and well-maintained supply chain management (SCM) system—one that can ensure an adequate supply of essential healthcare products to the patients who need them [21]. However, public healthcare supply chains are not only growing in size; they are also becoming more complex, with an increasing number of interconnected parts and feedback loops within the system [17]. Furthermore, public health supply chains face major challenges due to rising patient expectations and inefficiencies in supply chain operations. This has sparked interest in increasing public health supply chain efficiency and improving patient services [19]. Volatility and scarcity of information are also the major challenges in the healthcare supply chain that inhibit its capability to be responsive. To manage this demand variability, Akbarpour [20] highlights that contracts and enhanced procurement models, inventory routing, and vendor-managed inventory (VMI), as well as prepositioning supplies, will reduce the impacts of stockouts [20]. It seems that designing healthcare supply chain networks which account for unpredictability and the resulting uncertainties can assist managers in the decision-making process. In this paper we fill the abovementioned gaps by proposing a demand-driven supply chain management framework.

### 2.2. Demand-Driven Supply Chain Management

Practice in supply chain management emphasizes the importance of providing products and services that meet real market demand, using demand-driven supply chain management (DDSCM). This is a customer-activated pull system (CAPS). In a CAPS, the downstream supply chain partners place orders and upstream supply chain partners

in turn produce and deliver [22]. Information among supply chain partners flows in both ways, while products flow from the upstream supply chain partners to the downstream supply chain partners [23]. CAPS is more appropriate for products with high demand uncertainty and that are very costly [8]. In manufacturing and distribution, CAPS minimizes inventories, reduces costs, and enhances the ability of organisations to manage resources effectively. However, it is difficult to implement CAPS when lead times are long, making the response to demand information practically impossible [24]. Uncertain demand (e.g., sudden changes in product specifications, late orders, and changes to the volume of orders) makes it difficult to perform full delivery on time [25].

The CAPS in a healthcare supply chain context is when each health facility performs requisitions on the number of medicines it needs based on its past consumption and the status of stock. This is only possible using local information on consumption and demand [26]. The first classic pull system can be tracked back to a two-card Kanban system which was pioneered by Toyota. The two-card Kanban system implies that production on a workstation will start only when the succeeding station has requested or "pulled" parts, and not before [27]. A production pull system is conceptualized as a system which explicitly limits work-in-progress (WIP), as opposed to a production push system, which does not limit the amount of WIP [28]. Hull [27] likened push and pull systems to make-to-stock (MTS) and make-to-order (MTO), respectively, while Christopher et al. [11] used lean and agile supply chains to explain push and pull systems, respectively. Demand-driven production is therefore conceptualized as the synchronized execution of compliant production and logistics processes across a supply network to satisfy actual customer demand [29]. Consequently, DDSCM can be defined as extending the view of operations from a single organisation to the whole network. It focuses on the development of synergy along the whole network as opposed to internal optimization. For Ayers and Malmberg [30], an organisation can only transform from being "forecast-driven" to a "pull-driven" strategy on a continuum from being zero to 100% demand-driven with production/or inventory decisions that are entirely forecast-based to a situation in which orders are received prior to production/or replenishment. The goal is to manage uncertainties, volatility, and variability. Carbonara [31], suggests that a strategy of postponement in mitigating both supply and demand disruptions by considering the value of managerial flexibility to decide whether to exploit the strategy if and when disruptions occur and whenever product differentiation proves valuable based on the information available at that time. Supply chain disruptions may have a tremendous impact on organisations. Studies that have considered how to properly manage and reduce companies' exposure to supply chain disruptions have recognized the combination of both operational and strategic flexibility and redundancy as effective practices to cope with such risks [31]. This paper only focuses on operational and strategic flexibility under the context of DDSCM.

According to Bvuchete et al. [19], DDSCM can be defined by seven dimensions and associated capabilities as shown in Figure 1. These key dimensions include, technology, visibility, organisational alignment, collaboration, human resources, distribution management, and performance management. In this paper these seven dimensions and associated capabilities are used as a baseline for the development of the DDSCM network maturity mapping tool as well as to explain the concept of DDSCM as it applies to the healthcare industry.

### 2.2.1. Technology

Interoperable technology platforms make it possible to build demand-driven supply chains. They support seamless flow of information between all stakeholders in the supply chain network, thereby enabling organisations to deliver customer requirements cost-effectively and on time. A fast data-exchange platform can facilitate the exchange of data among supply chain partners [7,21]. Moreover, technology enhances the collection, analysis, and accessibility of near real-time demand and inventory information at different nodes in the supply chain, enabling informed planning and demand-based replenishment as opposed to forecast-driven replenishment [17]. Furthermore, ICT-based decision sup-

port tools coupled with human decision-making facilitates coordinated and collaborative environments for planning and decision-making [7]. Key requirements for a state-of-the-art information system include strategic direction and focus, platform and process integration, information coverage and availability, flexibility, adaptability, and information quality [19].

### 2.2.2. Visibility

Demand-based decisions can be enhanced by end-to-end supply chain visibility that is facilitated by technology, platform integration, and transparency [7]. Visibility can be interpreted as the intersection between information-sharing and information quality [19]. Information sharing is an optimization strategy aimed at improving supply chain coordination and integration [32], while information quality can be described using seven characteristics: relevance, timeliness, continuous flow, validity, accuracy, intelligibility, and usefulness [33,34]. End-to-end supply chain visibility enables organisations in the supply chain network to be responsive to volatile demand [17]. Inventory and demand visibility facilitates actual demand-based replenishment [19].

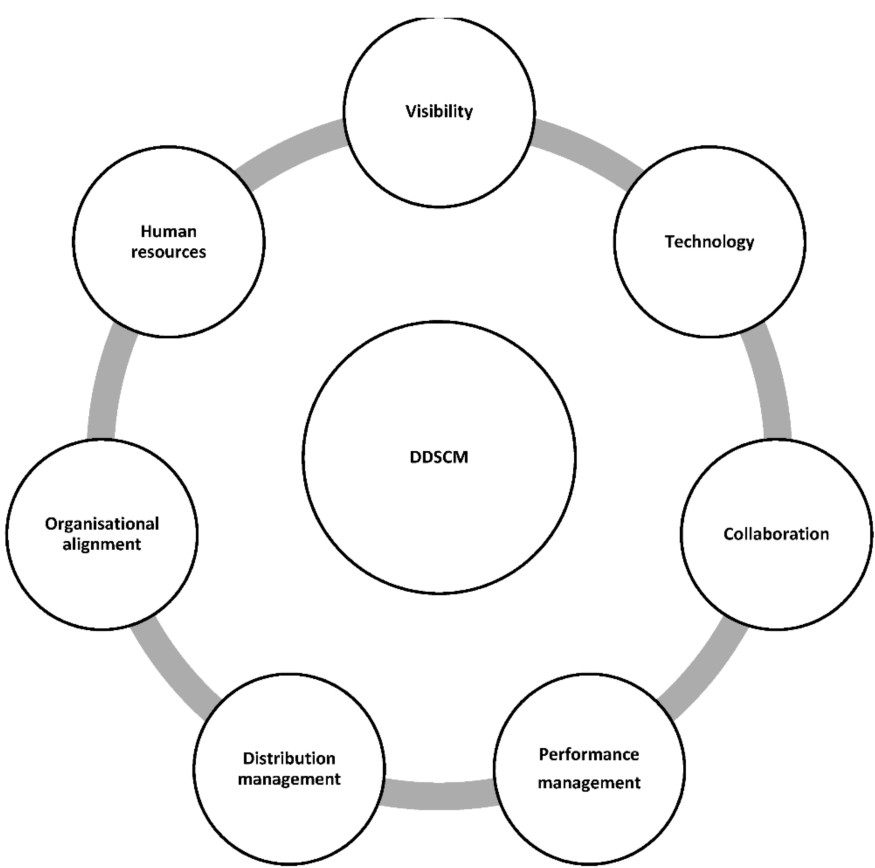

**Figure 1.** The proposed demand-driven supply chain management framework [19].

### 2.2.3. Organisational Alignment

The dimension of Organisational alignment entails the transformation process in which the organisational vision focuses on the "outside-in strategy" [29]. The outside-in strategy has the customer as the starting point of the supply chain [6]. The supply chain is then designed and aligned to customer needs. The first step is to understand the actual customer demand and consumption patterns. This will help in supply chain segmentation, product classification, the positioning of strategic decoupling points, and defining dynamic buffer profiles for these strategic decoupling points. Moreover, there is also a need for clearly outlining a vision for transition of the organisation from a "push-based" organisation into a "pull-based" organisation [9,19]. Lastly, the organisational culture, roles,

and responsibilities need to be in sync with the outlined vision [7]. However, for this to be successful, top management support is required [8].

### 2.2.4. Collaboration

The Collaboration dimension describes long-term partnership processes where stakeholders with common goals work closely together to achieve mutual benefits that organisations would not be able to achieve individually [35]. Commitment enables the formation of relationships and partnerships and subsequent information flows in a supply chain. Collaboration in a supply chain is also rooted in the concept of mutuality. Mutuality involves the sharing of supply chain risks, costs, and benefits among the supply chain partners [19]. Notably, joint collaboration planning actions influence the strength of the relationships and the use of inter-organisation information systems in a demand-driven supply chain [36]. This also requires that supply chain partners orchestrate decisions in supply chain planning and operations that optimise supply chain benefits. Joint planning is used to align supply chain plans and coordinate decisions on inventory replenishment, order placement, and order delivery [35].

### 2.2.5. Human Resources

The Human resources dimension describes the required skills, expertise, experience, and capacity of supply chain staff to manage and coordinate the supply chain [37]. To enhance this dimension, organisations need to invest in talent development through continuous training programmes that are certified by professional bodies. Since people are the drivers of supply chain innovation, mechanisms should be in place to capture the new ideas of supply chain staff and to move the ideas into action [19].

### 2.2.6. Distribution Management

Distribution management can be defined as an organisation's ability to facilitate the storage and flow of products to satisfy customer demand in a reliable and efficient way [38]. Distribution management capabilities involve best practices such as warehouse and inventory management, cross-docking, distribution planning and transportation management, customer order management and order fulfilment, and demand and supply planning [19].

### 2.2.7. Performance Management

Performance management entails measurement of supply chain performance. Supply chain performance measurements require a consistent and comparable holistic hierarchy of indicators, based on agreed upon strategies, performance targets and priorities [39]. Key metrics should be defined to monitor supply chain operations and evaluating supply chain performance. The standard supply chain metrics from the supply chain operations reference (SCOR) model (reliability, cost, responsiveness, and agility) are adapted in the DDSCM framework developed by Bvuchete et al. [19].

### *2.3. Maturity Models*

The concept of maturity models represents the stages through which processes and organisations progress as they are defined and improved [40]. There are five common characteristics that define maturity models. Firstly, maturity models can be characterised by their architecture. Two types of architectures exist: fixed-level and focus-area architecture. Fixed-level architecture is further classified into staged and continuous [41]. Secondly, maturity models can be classified into different types: maturity grids, Likert-like questionnaires, hybrid, and capability maturity model types [40]. Thirdly, they can be characterised by three common dimensions: process, technology, and people [30]. Fourthly, maturity models can be characterised by four traits. They have: (i) 3–6 maturity levels, (ii) a generic level descriptor, (iii) several dimensions, and (iv) each level description is based upon the activities associated with that level [40]. Lastly, they can be classified by their design purpose. They can be used as a tool to assess the "As-Is" situation (descriptive maturity model)

to guide the development of an improvement and control roadmap (To-Be) (prescriptive maturity model). Furthermore, a maturity model can be used as a benchmarking model for similar processes in other industries (comparative maturity model) [42]. A maturity model can also be viewed as a descriptive tool for the evaluation of strengths and weaknesses; it is a prescriptive instrument to help develop a guide (roadmap) for performance improvement, and it is a comparative tool to evaluate processes/organisations and compare them with the standards and best practices of other organisations, allowing them to implement external benchmarking [43].

Notably, the origins of maturity models can be traced back to the development of a quality maturity grid in 1979 [44]. The quality management grid inspired the development of a capability maturity model (CMM) by the United States of America Defence Software Engineering Institute. This is the most widely used maturity model concept representation [44]. The CMM provides a continuous software improvement path towards process capability [45], as shown in Figure 2. Ever since, several maturity models have been developed for different disciplines such as innovation, research and development, product and software development, supplier relationships and SCM [8]. In this study the CMM was used as a fundamental basis for the development of the network maturity mapping tool.

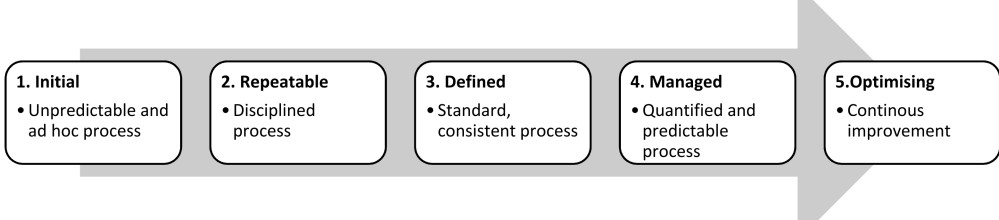

**Figure 2.** Capability maturity model representation [45].

The maturity model is a conceptual structure composed of parts that describe the development of a particular area of interest over time. Maturity increases with the improvement of the implemented practices and technologies used to manage the supply chain. A maturity model represents a set of structured levels of management capacity that characterize the performance of an organisation [43].

Immature organisations are characterized by improvisation in management, without establishing the required connections between the various knowledge areas. The appropriate level of maturity may vary depending on the available resources and the organisational needs. The adoption of improvement processes for the progression in maturity levels can be identified. The method employed to assess maturity is the application of questionnaires to determine the organisation's current maturity stage. Overall, the overarching objective of all maturity models is to improve the maturity of the organisations that use them, improving their processes [46].

### 2.4. Supply Chain Mapping

Supply chain mapping outlines the representation of linkages and partners of a supply chain. This ensures that the supply chain strategy conforms to the organisational strategy. It provides a complete, end-to-end picture of the supply chain [47]. Supply chain mapping also provides a viewpoint and representation of the supply chain network, its actors, and relationships between actors, as well as their associated processes [48].

Capability mapping captures relevant capabilities of the supply chain network [34]. This helps in the understanding of an organisation's supply chain, for the purposes of evaluating the current supply chain performance and contemplating the realignment of a supply chain [49]. Capabilities represent a structured set of building blocks that define an organisation—"what" an organisation does or has the capacity to do [49]. Further,

capability maturity mapping involves the assessment of the current state of operations and processes [47]. It also encompasses the determination of possible transition states and future mapping operations and processes as well as the development of actions/interventions and strategy to improve the current state of operations/processes [48]. Furthermore, stakeholder mapping and analysis (SMA) is the process of identifying all the stakeholders that may affect or be affected by the actions or decisions of a focal organisation [47]. SMA is a collaborative process of identifying, analysing, mapping, and prioritising stakeholder relevance. It also encompasses techniques for analysing the power, predictability, and interest of stakeholders [50]. In principle, stakeholders can be categorised into primary and secondary stakeholders [51]. Primary stakeholders include supply chain participants and organisations involved in the entire process of delivering the final product to its end consumer, i.e., from suppliers to end consumers. Secondary stakeholders are not directly involved in the organisation's primary activities [51]. Moreover, it is imperative to decide on the software tool that can be used to provide visual representation of supply chain mapping outcomes. This will allow a systems-level visual representation, helping management to identify the capabilities that need to be prioritised. This study utilised the Tableau software platform (TSP) that has been used in other studies related to supply chain mapping [47]. Lastly, network flow logic (NFL) outlines the three major flows in a public healthcare SCN: inventory, information, and finance [52].

## 3. Research Methodology

This section presents a detailed overview of the methodological steps followed in the development of the tool. A grounded theory-based approach was followed to develop the network maturity mapping tool, using the conceptual framework analysis (CFA) process proposed by Jabareen [53]. Data sources were mapped, and concepts identified, deconstructed, and categorised. The concepts were then integrated and synthesised into the network maturity mapping tool. The CFA process concluded with the evaluation and rethinking of the tool.

The tool development was evolutionary as shown in Figure 3. The first step involved adapting a preliminary framework by Bvuchete et al. [19] and associated capabilities extracted from literature. The preliminary framework consists of an inventory list of concepts that represent DDSCM, as shown in Figure 1. The framework was then evaluated in two stages: (E1) An evaluation of the concepts in the preliminary framework using semi-structured interviews with subject matter experts who have knowledge in DDSCM; and (E2) ranking interviews with subject matter experts that work directly in public healthcare supply chains.

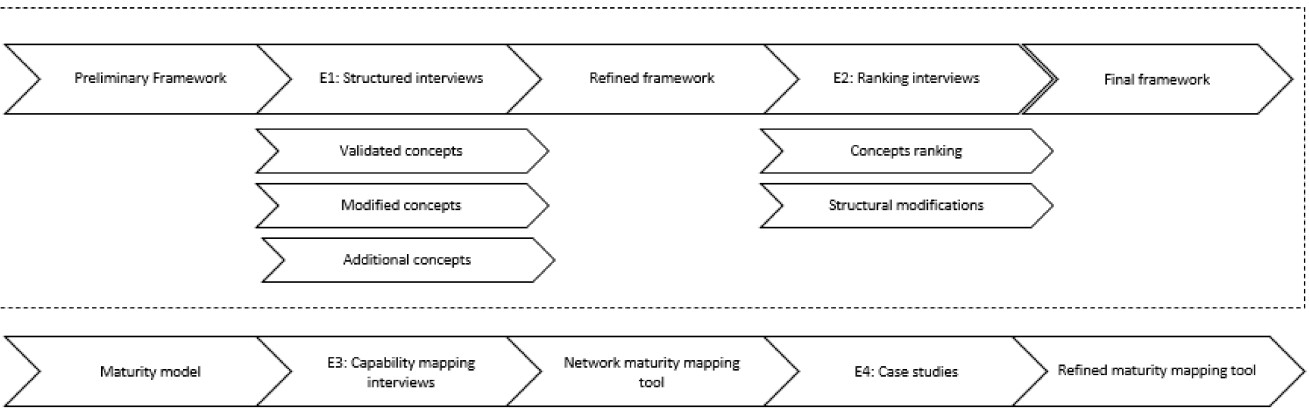

**Figure 3.** Methodological steps for the development of the network maturity mapping tool.

The second step was the development of a maturity model using the validated preliminary framework as a foundation. Furthermore, using capability mapping interviews (E3),

the maturity model later evolved into a NMMT which was applied in three case studies (E4). The detailed methodology is as shown in Table 1.

**Table 1.** Tool evaluation.

| Framework Evolution | Evaluation Stage | Outcomes of the Evaluation Stage |
|---|---|---|
| Preliminary framework | - Semi-structured interviews with 14 subject matter experts from both academia and industry, locally and internationally (E1). <br> - Purposive sampling was used in selecting the subject matter experts for this validation stage because the research needed to be validated by people who could provide information by virtue of their knowledge and experience | - Validate the concepts in the framework for credibility <br> - Modify concepts in the framework <br> - Test the framework for completeness and addition of new concepts |
| Preliminary framework | Ranking interviews with 9 subject matter experts (E2) and purposive sampling was used to select these subject matter experts | - Evaluation of the framework based on relevance, and useful in the context of public healthcare supply chains. This resulted in a final framework. |
| Maturity model | Capability mapping with 5 supply chain practitioners at different supply chain nodes (E3) (manufacturing, distribution, clinics, and hospitals) | - Capabilities in the maturity model are mapped to relevant supply chain nodes <br> - Capabilities profile |
| Network maturity mapping tool | Case studies on different supply chain nodes (E4) | - Evaluation of the network maturity mapping tool based on its applicability and validity on unique supply chain nodes in the public healthcare supply chain network. |

### 3.1. DDSCM Maturity Model Development

In line with recommendations from Fraser et al. [40], the DDSCM maturity model adapted a five-stage structure: Stage 1: Initial; Stage 2: Repeatable; Stage 3: Defined; Stage 4: Managed; and Stage 5: Optimised. Stage 1 represents unpredictable and ad hoc processes, while Stage 5 represents optimisation and continuous improvement of processes.

Organisations situate at any maturity stage based on the positioning and maturity of the seven dimensions outlined in the preliminary framework which include: organisation alignment, distribution management, collaboration, visibility, technology, human resource, and performance management. Figure 4 shows key aspects integrated into the maturity model.

After the development of the DDSCM maturity model, capability mapping interviews were performed to match the seven dimensions to relevant supply chain nodes which include manufacturing, distribution, hospitals, and clinics. Consequently, this led to the evolution of the maturity model into a network maturity mapping tool (Appendix A).

### 3.2. Capability Maturity Mapping

Finally, to test the applicability of the network maturity mapping tool, capability maturity mapping was performed on three case studies (manufacturing, distribution, hospitals/clinics). The aim was to explore, (i) the maturity of the dimensions and related capabilities, (ii) the impact of the dimensions and related capabilities on the performance of the organisation, and (iii) the effort required to implement and sustain the dimensions and related capabilities in the organisations. A Likert scale was used, as shown in Table 2.

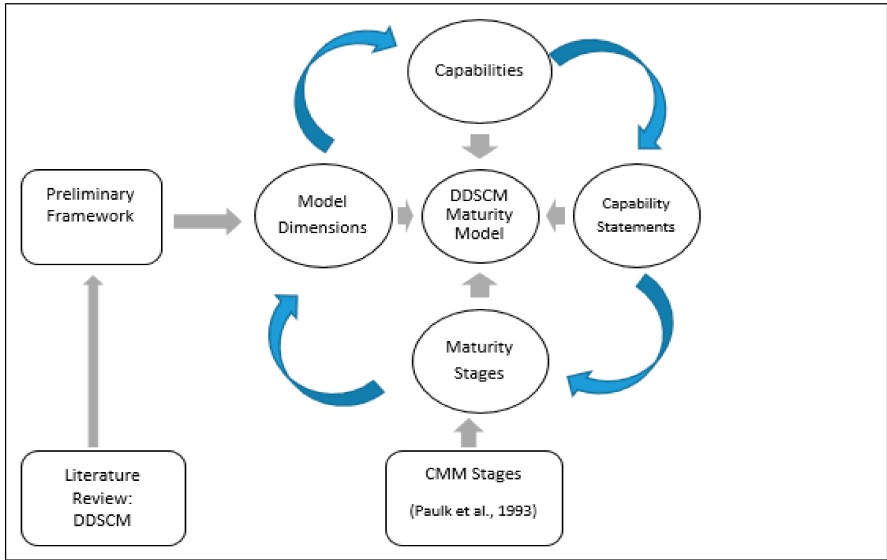

**Figure 4.** Key aspects integrated into the DDSCM maturity model.

**Table 2.** DDSCM capability mapping guidelines.

| Extent of Utilising a Capability $X_i$ | | Effort Required to Achieve Capability | | Impact of a Capability | |
|---|---|---|---|---|---|
| 0 | not at all | 1 | No effort | 1 | No impact |
| 1 | to a smaller extent | 2 | Some effort | 2 | Some impact |
| 2 | to some extent | 3 | Moderate effort | 3 | Moderate impact |
| 3 | to a moderate extent | 4 | Most effort | 4 | Most impact |
| 4 | To a larger extent | 5 | Extremely more effort | 5 | Extremely largest impact |

The capability maturity score (MS) can be determined for each capability using the average weighting approach as shown in the following Equation (1):

$$MS = \frac{\sum W_i X_i}{\sum W_i} \tag{1}$$

where $W_i = Particular\ Weight$ (each maturity level is given an equal weight).
$X_i = $ "*Particular value of the Extent of utilising a Capability*".

## 4. Results and Discussions

### 4.1. Validating the Preliminary Framework for Completeness (E1)

The dimensions and capabilities in the preliminary framework were validated with subject matter experts (SMEs). The purpose for this was to establish whether the preliminary framework was complete while capturing new concepts from SME based on actual practice. Capabilities that were emphasised by two SMEs and not contradicted by any of the SMEs were considered validated. Those capabilities that were described differently by SMEs but were like the capabilities in the preliminary framework were considered modified. New concepts that SMEs pointed out were added as shown in Table 3.

**Table 3.** E1, preliminary framework evaluation.

| Dimensions | Validation | Modification | Addition |
|---|---|---|---|
| Organisational alignment | • Strategic decoupling points (inventory positioning) <br> • Product classification <br> • Segmentation <br> • Buffer profiles and sizes <br> • Customer-oriented organisation (organisational vision) <br> • Leadership and management support | • Optimise strategic outsourcing activities <br> • Streamlined processes <br> • Supply chain cost measurement and management <br> • Alignment of supply chain metrics <br> • End-to-End alignment of supply chain goals and objectives | • Innovation to fusion all the DDSCM components into current supply chain processes |
| Distribution management | • Demand planning <br> • Warehouse inventory management <br> • Warehouse management system and technologies <br> • Warehouse layout reviews, picking and sorting <br> • Order management <br> • Expired product management <br> • Risk management | • Cross docking <br> • Flexible and responsive logistics systems <br> • Warehouse automation technologies <br> • Cycle counting and inventory audit | • Dynamic buffer adjustments <br> • Buffer stock modelling and calculations <br> • Usage behaviour <br> • Usage of industry 4.0 to capture demand |
| Collaboration | • Collaborative decision-making <br> • Strategic supplier relationship management <br> • Customer relationship management <br> • Risks and benefits sharing <br> • Mutual interdependence <br> • Information exchange <br> • Trust and commitment | • Resource sharing and scheduling <br> • Long-term contracts and service level agreements with suppliers <br> • Coordination among supply chain partners <br> • Supplier development <br> • Supplier performance measurement <br> • Collaboration for problem solving | N/A |
| Visibility | • Demand visibility <br> • Order visibility <br> • Visibility to relevant reports <br> • Lead-time visibility <br> • Forecast visibility <br> • Access to minimum order quantity (MOQ) and economic batch quantity information | • Buffer (inventory) visibility | N/A |
| Technology | • Integration of internal and external information sources (interoperable IT systems) <br> • Technological architecture that performs scenario analysis and buffer replenishment calculations <br> • Data capturing using technologies and sensors <br> • Data transmission and reporting through a common standardised supply chain platform <br> • Standardisation of data collection process | • Buffer monitoring and automatic order generation | • Simulation and route optimisation tool <br> • Supply chain control tower <br> • Big data analytics |
| Human resources | • Education, training, and certification of personnel by professional bodies and universities <br> • Skills, expertise, and competencies <br> • Innovation to solve complex SC problems <br> • Clear roles and responsibilities <br> • Incentives | • Development of rewarding strategies <br> • Management infrastructure that manage human resources <br> • Supportive structure for performance management <br> • Retention structure <br> • Good working environment and culture <br> • Systematic thinking ability of supply chain staff | • Workforce planning <br> • Organisational structure <br> • Human resources policies and plans <br> • Human resource information system <br> • Succession planning |

*4.2. Validating the Preliminary Framework for Relevance in the Healthcare Sector (E2)*

This validation phase involved ranking interviews to test the importance, relevance, and usefulness of the key dimensions and associated capabilities in the public healthcare sector as shown in Figure 5. Semi-structured interviews were conducted with nine subject matter experts specifically in the field of healthcare supply chains.

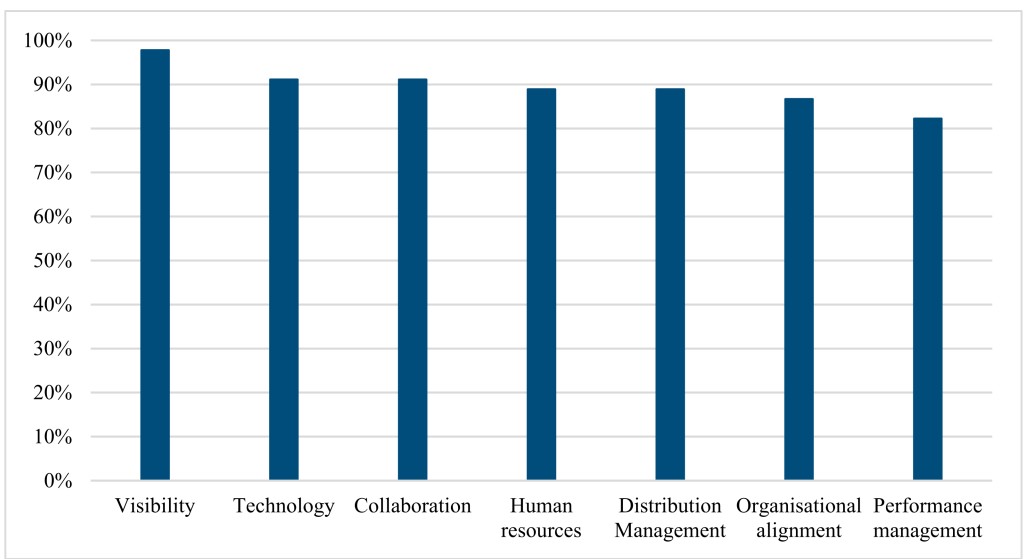

**Figure 5.** Importance and usefulness of key DDSCM dimensions in healthcare supply chains.

It can be noted in Figure 5 that the most important dimension in public healthcare supply chains is Visibility (97.8%) which can be enhanced by the Technology dimension (91.1%) and Collaboration (91.1%). The importance of collaboration in healthcare supply chain emphasizes the need of a network view and alignment. Next, the importance of both Human resources and Distribution management dimensions in public healthcare supply chain is very high, at 88.9%. Organisational alignment and Performance management dimensions have values of 86.7% and 82.2% respectively.

### 4.3. Capability Mapping: Matching Dimensions to Supply Chain Nodes (E3)

Different supply chain nodes in the public healthcare SCN are characterised by unique supply chain processes. This implies that, not all dimensions and related capabilities in the validated framework are applicable to all the supply chain nodes. Capability mapping involved mapping the capabilities in the framework to different supply chain nodes in the healthcare supply chain network.

The process of matching the dimensions and their associated capabilities involved semi-structured interviews with five SME at the unique supply chain nodes. The output from the capability mapping interviews was a capabilities profile as shown in Table 4. The capabilities profile outlines the alignment of DDSCM capabilities with regard to the unique supply chain nodes.

**Table 4.** Key capability mapping profiles (public healthcare supply chain in South Africa).

| Dimensions | ID# | Capability | Manufac-turing | Central Medicines Stores | Hospitals | Clinics | Department of Health | 3PLs |
|---|---|---|---|---|---|---|---|---|
| Distribution Management | 3 | Risk Management | ✓ | ✓ | ✓ | ✓ | ✗ | ✗ |
| | 4 | Demand Planning | ✓ | ✗ | ✗ | ✗ | ✓ | ✗ |
| | 5 | Inventory Positioning | ✓ | ✗ | ✗ | ✗ | ✓ | ✗ |
| | 6 | Supply Planning | ✓ | ✓ | ✗ | ✗ | ✓ | ✗ |
| | 7 | Distribution Planning | ✓ | ✓ | ✗ | ✗ | ✓ | ✗ |
| | 8 | Warehouse Operations (Inventory Management) | ✓ | ✓ | ✓ | ✓ | ✗ | ✗ |
| | 9 | Warehouse Visibility and Automation | ✓ | ✓ | ✓ | ✓ | ✗ | ✗ |
| | 10 | Order Management | ✓ | ✓ | ✓ | ✓ | ✗ | ✗ |
| | 11 | Logistics and transport Flexibility | ✗ | ✗ | ✗ | ✗ | ✗ | ✓ |
| | 12 | Expiry Management | ✓ | ✓ | ✓ | ✓ | ✗ | ✗ |
| Visibility | 13 | Information Sharing and Information Quality | ✓ | ✓ | ✓ | ✓ | ✗ | ✓ |
| | 14 | Demand and Inventory Visibility | ✓ | ✓ | ✓ | ✓ | ✗ | ✗ |
| | 15 | Order Visibility | ✓ | ✓ | ✓ | ✓ | ✗ | ✗ |
| Technology | 16 | Information Technology Infrastructure | ✓ | ✓ | ✓ | ✓ | ✗ | ✗ |
| | 17 | Data Transmission and Reporting | ✓ | ✓ | ✓ | ✓ | ✗ | ✗ |
| | 18 | Data Capturing | ✓ | ✓ | ✓ | ✓ | ✗ | ✗ |
| Collaboration | 19 | Relationships | ✓ | ✓ | ✓ | ✓ | ✗ | ✗ |
| | 20 | Interdependence and resource sharing | ✓ | ✗ | ✗ | ✗ | ✗ | ✗ |
| | 21 | Decision-Making | ✓ | ✓ | ✓ | ✓ | ✗ | ✗ |
| Performance management | 22 | Key Performance Indicators | ✓ | ✓ | ✓ | ✓ | ✗ | ✗ |
| | 23 | Metrics | ✓ | ✓ | ✓ | ✓ | ✗ | ✗ |
| Human Resources | 24 | Policies and plans | ✓ | ✓ | ✓ | ✓ | ✗ | ✗ |
| | 25 | Roles and responsibilities | ✓ | ✓ | ✓ | ✓ | ✗ | ✗ |
| | 26 | Innovation | ✓ | ✓ | ✓ | ✓ | ✗ | ✗ |
| | 27 | Incentives and Working environment | ✓ | ✓ | ✓ | ✓ | ✗ | ✗ |
| | 28 | Training and Education | ✓ | ✓ | ✓ | ✓ | ✗ | ✗ |
| Organisational Alignment | 29 | Management Support | ✓ | ✓ | ✗ | ✗ | ✗ | ✗ |
| | 30 | Organisational Vision | ✓ | ✓ | ✗ | ✗ | ✗ | ✗ |
| | 31 | Streamlined Processes | ✓ | ✓ | ✓ | ✓ | ✗ | ✗ |
| | 32 | Alignment of Objectives | ✓ | ✓ | ✗ | ✗ | ✗ | ✗ |
| | 33 | Cost Management | ✓ | ✓ | ✓ | ✓ | ✗ | ✗ |
| | 1 | Product Classification | ✓ | ✗ | ✗ | ✗ | ✓ | ✗ |
| | 2 | Segmentation | ✓ | ✗ | ✗ | ✗ | ✓ | ✗ |

*4.4. Capability Maturity Mapping: Case Studies (E4)*

This section summarises results obtained from the application of the network maturity mapping tool in case studies at three different nodes in the public healthcare supply chain network in South Africa (manufacturer, provincial pharmaceutical warehouse, and hospital, as shown in Table 5).

Figure 6 provides an overview of the maturity-impact mapping outcomes, while Figure 7 outlines the outcomes for the maturity-effort mapping from the case studies. The capabilities in quadrant I deserve to be given first priority since these capabilities have low maturity but have significant impact on the organisational performance. Second priority is given to capabilities in quadrant II, followed by quadrant III. The capabilities that will be given last priority are the ones in quadrant IV.

**Table 5.** Profiles of the three (3) cases in South Africa.

| Supply Chain Node | Stakeholders | Description |
|---|---|---|
| Manufacturer | Pharmaceutical Company | A science-led global healthcare company that aims to improve the health of people globally. It focuses on the research and development of products that include pharmaceuticals, vaccines, and consumer health products. |
| Distribution | Central Medicines Stores | Provincial pharmaceutical distributor and wholesaler for the Western Cape province. Procure, warehouse, and distribute drugs. |
| Customer | Hospital | The hospital is one of Cape Town's premier tertiary academic hospitals and was opened in 1938. It provides outstanding tertiary and quaternary care for patients of the Western Cape and beyond. This hospital operates in the Cape Town Central Health District of the Metro Region and is owned and supported by the Western Cape Department of Health (DoH). |

*4.5. Discussion of Case Study Results*

The purpose of capabilities maturity mapping is to identify capabilities that organisations should prioritise when they are designing interventions that will improve not only the performance of their organisation but also the entire supply chain network. Management can then commit resources and investments towards enhancing the identified capabilities. After putting into place interventions and controls to ensure the continuous improvement of capabilities, management should then evaluate the impact of the interventions on an annual basis. Capability maturity mapping exercises should be performed at a predetermined frequency. A continuous review of the capabilities over the years (twice a year) and successive interventions will facilitate organisations to progress towards advanced maturity stages of DDSCM.

From the case studies it was observed that different organisations representing different supply chain nodes have DDSCM capabilities at different maturity levels. This highlights opportunities for the improvement of these capabilities. Priorities should be placed on immature DDSCM capabilities, which have high impact on the performance of the supply chain node and require minimum effort for implementation and sustaining.

Furthermore, when comparing the maturity of capabilities between manufacturing, distribution, and the hospital, it was noted that the hospital has capabilities at lower maturity levels, while the pharmaceutical manufacturer has capabilities at high maturity levels. From the case study results presented in Figure 6, one could deduce that the higher the impact of the DDSCM capability on the organisational performance, the greater the maturity of the capability (a higher extent to which a capability is used). Therefore, our first hypothesis can be formed:

**Hypothesis 1 (H1).** *There is a positive correlation between capability maturity and the impact of the capability on organisational performance.*

Looking at Figure 7, no relationship between the effort required to implement/sustain capability and the maturity level of that capability could be established. This leads to our second hypothesis:

**Hypothesis 2 (H2).** *There is no relationship between the effort required to implement/sustain a capability and the maturity level of that capability.*

However, future work should look at validating these two hypotheses with multiple case studies. Table 6 highlights the different priorities that should be placed by the unique organisations in the public healthcare SCN in the case studies. The priorities listed are not due to current low maturity per se, but are focussed on where the node can achieve the highest next improvement with the least effort.

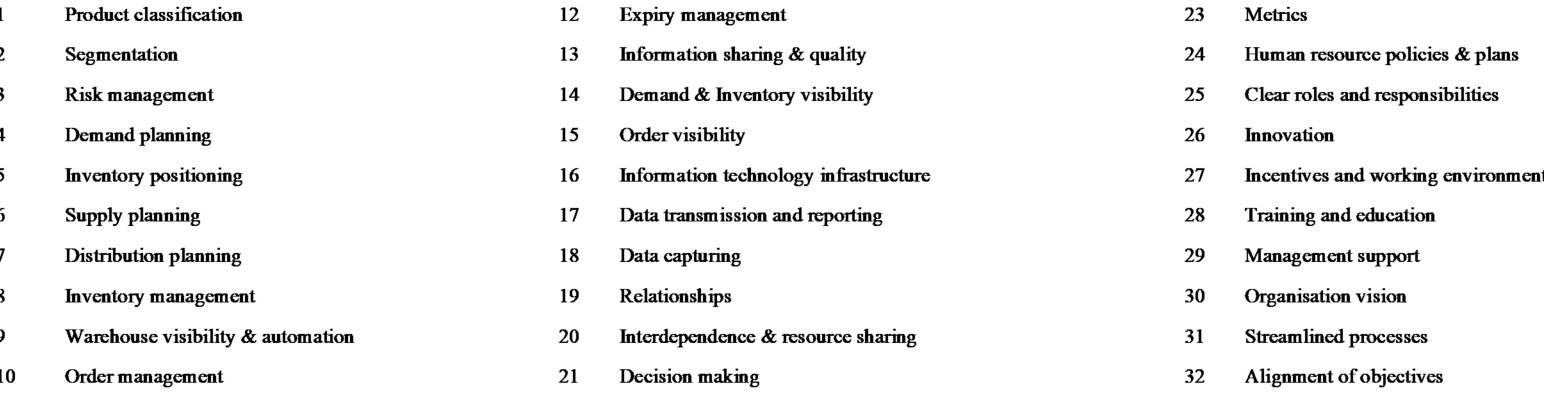

**Figure 6.** Maturity—impact mapping.

| | | | | | | |
|---|---|---|---|---|---|---|
| 1 | Product classification | 12 | Expiry management | 23 | Metrics | |
| 2 | Segmentation | 13 | Information sharing & quality | 24 | Human resource policies & plans | |
| 3 | Risk management | 14 | Demand & Inventory visibility | 25 | Clear roles and responsibilities | |
| 4 | Demand planning | 15 | Order visibility | 26 | Innovation | |
| 5 | Inventory positioning | 16 | Information technology infrastructure | 27 | Incentives and working environment | |
| 6 | Supply planning | 17 | Data transmission and reporting | 28 | Training and education | |
| 7 | Distribution planning | 18 | Data capturing | 29 | Management support | |
| 8 | Inventory management | 19 | Relationships | 30 | Organisation vision | |
| 9 | Warehouse visibility & automation | 20 | Interdependence & resource sharing | 31 | Streamlined processes | |
| 10 | Order management | 21 | Decision making | 32 | Alignment of objectives | |

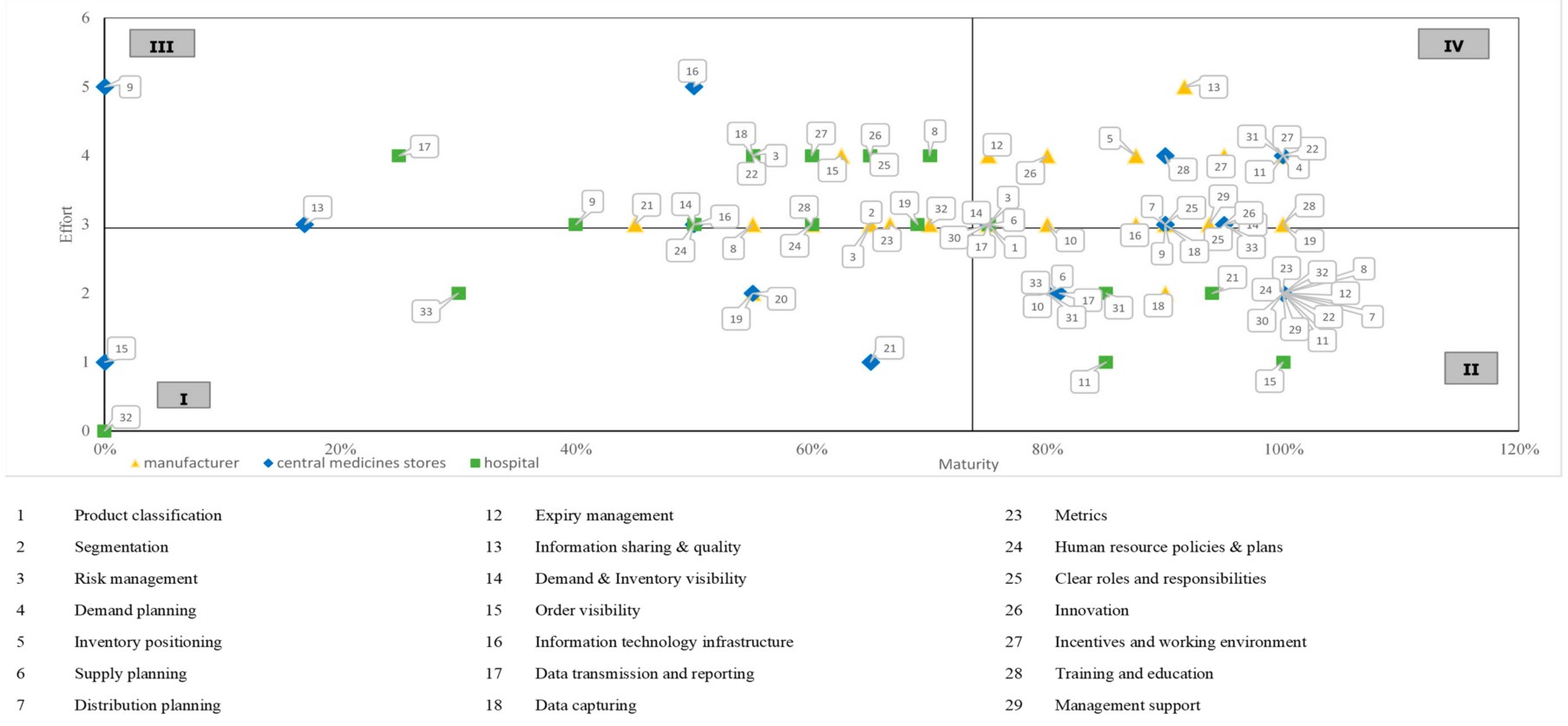

**Figure 7.** Maturity—effort mapping.

| | | | | | |
|---|---|---|---|---|---|
| 1 | Product classification | 12 | Expiry management | 23 | Metrics |
| 2 | Segmentation | 13 | Information sharing & quality | 24 | Human resource policies & plans |
| 3 | Risk management | 14 | Demand & Inventory visibility | 25 | Clear roles and responsibilities |
| 4 | Demand planning | 15 | Order visibility | 26 | Innovation |
| 5 | Inventory positioning | 16 | Information technology infrastructure | 27 | Incentives and working environment |
| 6 | Supply planning | 17 | Data transmission and reporting | 28 | Training and education |
| 7 | Distribution planning | 18 | Data capturing | 29 | Management support |
| 8 | Inventory management | 19 | Relationships | 30 | Organisation vision |
| 9 | Warehouse visibility & automation | 20 | Interdependence & resource sharing | 31 | Streamlined processes |
| 10 | Order management | 21 | Decision making | 32 | Alignment of objectives |
| 11 | Logistics and transport flexibility | 22 | Key performance indicators | 33 | Cost management |

**Table 6.** Priorities that need interventions.

| Node Type | Priority 1 | Priority 2 | Priority 3 | Priority 4 | Priority 5 |
|---|---|---|---|---|---|
| Manufacturer | **21.** Decision-making | **8.** Inventory management | **20.** Interdependency and resource sharing | **15.** Order visibility | **24.** Human resource policies and plans |
| Central Medicines Stores | **15.** Order visibility | **9.** Warehouse visibility and automation | **13.** Information sharing and quality | **14.** Demand and inventory visibility | **16.** Information technology infrastructure |
| Hospital | **17.** Data transmission and reporting | **33.** Cost management | **9.** Warehouse visibility and automation | **24.** Human resource policies and plans | **16.** Information technology infrastructure |

The case studies indicate that decision-making is the area in which a more concentrated effort is required at the manufacturing node. This entails joint planning and decision-making [54]. The decision-making process in the supply chain is facilitated by a supply chain control dashboard [54]. The level of synchronisation in the decision-making process is a key element in building and maintaining mutual partnerships [35].

In the central medicine stores (distribution), efforts to attain higher stages of DDSCM maturity should be concentrated first on ensuring order visibility through an integrated information technology infrastructure. This will enable tracking and tracing of the status of orders in the supply chain pipeline. The capability of warehouse (pharmacy) visibility and automation appears to be a priority for central medicine stores and the hospital.

It was also observed that there is a lack of information sharing and quality between the medicine manufacturer and the central medicine stores as well as the hospital. Hence, information technology-enabled solutions such as a supply chain dashboard that has the capabilities of data collection, data processing, modelling, and the communication and visualization of relevant information for the purposes of collaborative advanced decision support and development of alternative demand and supply scenarios are of vital importance to all the supply chain nodes. Notably, the results are very specific to organisations that were involved in the testing of the applicability of the tool and thus are not to be seen as indicative of all organisations across the public healthcare SCNs in South Africa. Thus, this is a unique result at this point in time for these organisations.

The main contribution of the practical case study approach is that it outlines the validity and applicability of the maturity mapping tool in the healthcare sector, and also provides insights from actual decision-makers for healthcare supply chain management. The results show that the tool can be used to assess DDSCM processes as well as to guide the improvement roadmap towards best practice. This study combined three research domains: healthcare, DDSCM, and maturity models, to create a theoretical contribution to the literature on healthcare supply chain optimization and the diversified field of maturity models.

## 5. Gap Analysis and Contribution

Designing healthcare supply chain networks which account for unpredictability and the resulting uncertainties can assist managers in the decision-making process. This paper focuses on the use of demand-driven supply chain management operating practice in the healthcare industry. The healthcare supply chain is a complex adaptive ecosystem where low forecast accuracy and high demand volatility result in an increase in stockouts, operational inefficiencies, and poor health outcomes, with a significant increase in supply chain costs. To cope with these challenges the authors analysed the implementation of DDSCM practices. Various contributions have been made in this paper. (i) Several efforts have been made to evaluate and improve individual performance with regard to the organisation's suppliers, distributors, and customers, but it appears there is lack of

network-level studies in the healthcare sector. (ii) This paper involves a combination of three research areas: the healthcare supply chain, maturity models, and DDSCM, with a contribution to the body of research and the public healthcare sector. (iii) Here, the tool is applied to a specific industry (the public healthcare sector in SA). (iv) A systems approach is used, focusing on the entire network and providing another contribution to both the public healthcare sector and DDSCM. (v) Finally, this study involves the design of the network maturity tool for assessment and as a guideline for healthcare supply chain optimization. The tool implemented using real case studies, from which useful managerial insights are derived as the main results of the study.

## 6. Management Implications

This study provides some insights from a managerial perspective as well. In this regard, the approach used in this study may be interesting for both organisational (node)-level managers as well as regional (district) and national (multi-tier) managers, as this type of analysis illustrates an extensive picture of the maturity of processes in the healthcare SCN with regards to DDSCM. Organisational managers can focus on improving individual processes, while the national managers can use the results for benchmarking purposes across multiple tiers in SCNs. This will enhance informed decision-making at the national level with regard to a series of improvement projects in SCNs. The tool equips managers with the ability to identify "best-in-class" nodes for a specific type (i.e., clinics), and then they can transfer learning from this "best-in-class" node to other nodes. However, the authors recommend that this approach first be piloted starting from downstream (health facilities) then moving up the chain towards the manufacturer. For instance, the intended new technology can be tested to show how it improves maturity at one/a few nodes, and then once proven, rolled out to other nodes of the same type. On one hand this approach demands the alignment of leadership, culture, customers (healthcare facilities), and operational performance across the entire SCNs. If organisations in public healthcare SCNs could provide everyone in the supply chain with accurate data regarding demand, inventory, and lead times, then organisations would be less likely to make decisions that lead to erratic swings in inventory. On the other hand, if suppliers can see actual demand from downstream, they are less likely to overreact to small variations. This visibility is accelerated by technologies that automate, digitize data, and connect every function within an organisation and every layer of SCNs. Finally, truly reaping the benefits of DDSCM requires that organisations make sure their operational processes are agile, adaptable, and aligned.

## 7. Conclusions

The healthcare supply chain is a complex adaptive ecosystem where low forecast accuracy and high demand volatility result in an increase in stockouts, operational inefficiencies, poor health outcomes, and a significant increase in supply chain costs. To cope with these challenges the authors analysed the implementation of DDSCM practices.

This paper presented a network-maturity mapping tool to assist unique supply chain nodes in public healthcare SCNs in assessing their current maturity level regarding DDSCM processes. Furthermore, the tool provides guidelines to set interventions to progress towards higher stages of maturity. The tool supports a journey of transformation of organisational processes. The key DDSCM domains include technology, visibility, human resources management, collaboration, organisational alignment, performance management, and distribution management.

Key insights from practical application of the tool in actual settings show that collaboration across all nodes in the public healthcare SCN is in its infancy. Moreover, ensuring supply chain visibility in the SCN is also posing a challenge. However, technical approaches involving data analytics support this DDSCM approach. These technical approaches also involve the successful assimilation of information systems to enable supply chain integration and supply chain visibility. Having the right technologies only brings some success,

and relationships along the supply chain are equally important. They facilitate the sharing of real-time planning, management, and execution information to enable a seamless flow of demand signals between supply chain partners. Consequently, collaboration is key to DDSCM, and requires participating organisations to respond to changes in demand, particularly when demand is volatile and unpredictable. The study underlines that clear agreements and collaboration between all players involved in the public healthcare supply chain can help solve the problems caused by lack of supply chain visibility, which has a significant impact on the availability of medicines at healthcare facilities.

Like any other study, this study also has its limitations. DDSCM is a relatively new approach, and for this reason there is little published data in this field and much confusion regarding key enablers. In addition, few studies found in the literature outline DDSCM from a functional perspective within a single organisation. The present work aims to go a step further to analyse the DDSCM from an end-to-end supply chain perspective and focus on the alignment of all supply chain partners towards customer demand. This has been a significant contribution of this study. The limitations of this study include a limited number of case studies performed to validate the applicability of the network maturity mapping tool. However, future studies should explore multiple case studies at different supply chain nodes.

It is also vital to clearly point out that the tool is biased towards people with supply chain experience. There is a higher risk that people without supply chain experience will interpret the contents of the tool wrongly, resulting in the tool giving assessment outcomes that are biased. Thus, implementation should take place with the necessary care and management effort.

Lastly, the healthcare supply chain community has been provided with a tool that can be used to aid decision-making when dealing with demand volatility and supply chain complexity. The designed tool is a major contribution to practice because it provides a stepwise approach to supply chain optimization. Furthermore, there is a major contribution to literature as the research combines different fields of research, healthcare, DDSCM, and maturity models to create content that can be applied in the context of healthcare. The applicability of the tool has been rigorously tested in the case study, with insights from experts in the field. Using this tool fosters continuous improvement in healthcare.

**Author Contributions:** M.B.: Conceptualization, methodology, validation, analysis, and editing. S.S.G.: Conceptualization, methodology, validation, analysis, and editing. J.v.E.: Conceptualization, methodology, validation, analysis, and editing. All authors have read and agreed to the published version of the manuscript.

**Funding:** This research received funding from GlaxoSmithKline (GSK).

**Institutional Review Board Statement:** Not applicable.

**Informed Consent Statement:** Written informed consent has been obtain from the participants of this study to publish this paper.

**Data Availability Statement:** All sources have been provided in the manuscript.

**Acknowledgments:** The support of GlaxoSmithKline (GSK) is highly noted but conclusions reached are of the authors.

**Conflicts of Interest:** The authors declare no conflict of interest.

## Appendix A

**Table A1.** A Network Maturity Mapping Tool for Demand Driven Supply Chain Management.

| Key Supply Chain Nodes and Capabilities Profile | | | Maturity Model Capabilities | Maturity Model Stages | | | | |
|---|---|---|---|---|---|---|---|---|
| Manufacturing | Distribution | Clinics/Hospitals | | Initial Unpredictable and ad hoc Process | Repeatable Disciplined Processes | Defined Quantified and Predictable Process | Managed Quantified and Predictable Process | Optimised Continuous Improvements |
| **Organisational Alignment** | | | | | | | | |
| ✓ | ✗ | ✗ | Product classification | No product categorisation strategy in place | Specific products and their related service levels (quality, cost) are identified | Product categorisation using the (DWV3) framework | Segment products into innovative or functional products | Alignment of product categories with different supply chain strategies |
| ✓ | ✗ | ✗ | Segmentation | No segmentation plan in place | Segmentation analysis specifying supply chain goals and scope | Established customer and supplier segments with specified service levels | Customised response to the established segments | Segmentation strategy reviewed and updated yearly |
| ✓ | ✗ | ✗ | Inventory positioning | Staged inventories caused by failure to integrate | Cross docking and direct deliveries to achieve delivery efficiencies | Strategic decoupling points positioning | Strategic decoupling points protected by buffer profiles and replenished based on actual demand | Dynamic adjustments of buffer profiles and sizes based on events and seasonality |
| ✓ | ✓ | ✗ | Management support | No management support for supply chain processes and resource allocation | Developed digital supply chain strategies to replace unnecessary inventory movements | Development of holistic demand-driven supply chain strategy (demand pull replenishment) | A top management support demand-driven supply chain management approach and support of the building of adaptive demand-based networks | A culture of continuous improvement is entrenched |
| ✓ | ✓ | ✗ | Organisational vision | Little understanding of the value of actual customer demand data (forecast-driven organisation) | Understanding of actual customer demand and translation of that into strategies and plans to satisfy the actual demand | Organisational vision does clearly define customer orientation and demand-pull replenishment as a core competency | Customer-pull-driven fulfilment as well as outsourcing all non-core supply chain activities to third-party logistics | A customer-focused, collaborative culture is firmly in place (demand-driven organisation) |

**Table A1.** *Cont.*

| Key Supply Chain Nodes and Capabilities Profile | | | Maturity Model Capabilities | Maturity Model Stages | | | | |
|---|---|---|---|---|---|---|---|---|
| Manufacturing | Distribution | Clinics/Hospitals | | Initial Unpredictable and ad hoc Process | Repeatable Disciplined Processes | Defined Quantified and Predictable Process | Managed Quantified and Predictable Process | Optimised Continuous Improvements |
| **Organisational Alignment** | | | | | | | | |
| ✓ | ✓ | ✓ | Streamlined processes | Basic definition of supply chain processes and stock management processes | Processes are well-defined and documented | Supply chain standards to assure customer quality standards are met effectively | Organisational strategy integrates demand processes and supply processes | Processes are optimised and continuously improved to ensure agility |
| ✓ | ✓ | ✗ | Alignment of objectives | Forecast-driven supply chain strategy for all product types | A demand-driven supply chain strategy is defined | A comprehensive demand-driven supply chain strategy is implemented | End-to end supply chain goals and objectives are collaboratively developed and supply chain partners perceive that their organisation goals are satisfied by accomplishing supply chain objectives. | Continuous review and update of the demand-driven supply chain strategy |
| ✓ | ✓ | ✓ | Cost management | Identified need to track supply chain cost | All supply chain costs are identified | Supply chain costs are measured, controlled, and managed | Financial deviations from target are actively managed | Full visibility of supply chain costs amongst supply chain partners through the digitalised supply chain platform |
| **Visibility** | | | | | | | | |
| ✓ | ✓ | ✓ | Information sharing and information quality | Information sharing is not a priority | Information sharing in functional silos | Information accessibility across the entire organisation | Transparency across demand, inventory, and capacity information amongst supply chain stakeholders in the network | Information availability, accessibility, and usability from all actors on a supply chain platform driving joint planning |
| ✓ | ✓ | ✓ | Demand and inventory visibility | Demand and inventory visibility (<20%) | Visibility on inventory and consumption patterns, minimum order quantity and economic batch quantity (<40%) | Market intelligence to identify which products are wanted by which customers, and lead-time visibility | Inventory and demand information is shared across the supply chain to provide visibility to all echelons (<80%) | Real-time visibility on consumption patterns, inventory status, and scheduled deliveries (<100%) shared across the entire pipeline and is used to make informed decisions |

**Table A1.** *Cont.*

| Key Supply Chain Nodes and Capabilities Profile | | | | Maturity Model Stages | | | | |
|---|---|---|---|---|---|---|---|---|
| Manufacturing | Distribution | Clinics/Hospitals | Maturity Model Capabilities | Initial Unpredictable and ad hoc Process | Repeatable Disciplined Processes | Defined Quantified and Predictable Process | Managed Quantified and Predictable Process | Optimised Continuous Improvements |
| **Visibility** | | | | | | | | |
| ✓ | ✓ | ✓ | Order visibility | Customer order visibility and tracking (<20%) | Customer order visibility and tracking (<40%) | Customer order status visibility and tracking (<60%) | Order transaction and movement is visible to supply chain partners and customers (<80%) | Real-time status throughout the order pipeline, online real-time order configuration and updates through the supply chain platform |
| **Technology** | | | | | | | | |
| ✓ | ✓ | ✓ | Information technology infrastructure | Legacy technologies that are not integrated with other technology systems | Basic integration of information management systems in an organisation | Fast-data exchange platform (robust technology infrastructure) that can share inventory data in real-time among all participants in the supply chain | New technologies and sophisticated analytics to make the supply chain more responsive to customer demand and alerts in case of unexpected deviations | The platform necessitates the ability to simulate supply alternatives, carry out risk and benefit analysis in real-time |
| ✓ | ✓ | ✓ | Data transmission and reporting | IT architecture produces static reports and the reports are distributed in paper format | IT supports static reports with graphical data and reports are distributed digitally | IT architecture supports dynamic data navigation and reports are distributed automatically and digitally | IT architecture supports dynamic statistical analysis and reports are directly and constantly available to relevant supply chain partners | IT architecture support dynamic scenario analysis and reports are directly and constantly available to relevant supply chain partners |
| ✓ | ✓ | ✓ | Data capturing | Manual capturing and reporting of medicines consumption data | Electronic solutions to strengthen data collection and reporting | Most master data consistently defined but not entirely harmonised throughout the organisation | Master data proactively managed internally but not externally | Master data consistently defined and harmonised throughout the supply chain |

**Table A1.** *Cont.*

| Key Supply Chain Nodes and Capabilities Profile | | | Maturity Model Capabilities | Maturity Model Stages | | | | |
|---|---|---|---|---|---|---|---|---|
| Manufacturing | Distribution | Clinics/Hospitals | | Initial Unpredictable and ad hoc Process | Repeatable Disciplined Processes | Defined Quantified and Predictable Process | Managed Quantified and Predictable Process | Optimised Continuous Improvements |
| **Collaboration** | | | | | | | | |
| ✓ | ✓ | ✓ | Relationships | Formalised and rationalised framework for collaboration in place | Supply chain actors value collaboration (internally) | Great efforts made on building good relationships with suppliers (supplier relationship management) and strategic alliances as opposed to adversarial relationships | Partnerships and advanced relationships with suppliers and customers. | Supplier development, long-term contracts, and supplier sustainability gap analysis to identify the constraints faced by suppliers |
| ✓ | ✗ | ✗ | Interdependence and resource sharing | Commitment by supply chain partners with respect to collaborative resource investment and resource pooling | Pooling and sharing resources together with other supply chain partners to reduce the supply chain risk due to supply chain disruptions | Commitment agreements signed for collaborative resource investment and sharing with supply chain partners | Vertical and horizontal dependences within and between chains, sharing risk and benefits. | Resource sharing and pooling included in organisational strategy and are continually reviewed |
| ✓ | ✓ | ✓ | Decision-making | Joint knowledge creation and innovation with supply chain partners (internally) | Collaborative setting and alignment of KPIs with other supply chain partners | Data is used as an input to decision-making | Collaborative decision-making and problem solving with other supply chain partners driven by available data (externally) | End customer consumption drives decisions along the chain |
| **Performance management** | | | | | | | | |
| ✓ | ✓ | ✓ | Key performance indicators | Basic performance management plans are defined and imbedded in the organisation with indicators | ICT-based measurement systems | Regular monitoring, tracking and updating of KPIs against goals and there is transparency to create an environment of continuous improvement | KPIs support managers to assess the overall supply chain performance, diagnose problems, and plan actions progressively | Root cause analysis and corrective actions consistently taken into consideration to improve performance based on monitoring and evaluation results |

Table A1. *Cont.*

| Key Supply Chain Nodes and Capabilities Profile | | | Maturity Model Capabilities | Maturity Model Stages | | | | |
|---|---|---|---|---|---|---|---|---|
| Manufacturing | Distribution | Clinics/Hospitals | | Initial Unpredictable and ad hoc Process | Repeatable Disciplined Processes | Defined Quantified and Predictable Process | Managed Quantified and Predictable Process | Optimised Continuous Improvements |
| **Performance management** | | | | | | | | |
| ✓ | ✓ | ✓ | Metrics | Key metrics not defined in the organisational strategy (reliability, stability, speed/velocity, responsiveness) | Need recognised for defining key metrics (reliability, stability, speed/velocity, responsiveness) | Key metrics defined (reliability, stability, speed/velocity, responsiveness) | Key metrics measured and quantified (reliability, stability, speed/velocity, responsiveness) | Controls in place to improve supply chain (reliability, agility, stability, speed/velocity and responsiveness) |
| **Human resources** | | | | | | | | |
| ✓ | ✓ | ✓ | Policies and plans | A strategic plan that addresses human resource requirements for supply chain functions and personnel is authored | The supply chain organisation structure adequately supports supply chain functions | Succession plans are updated annually and used to inform recruiting, workforce development, and other staffing decisions | A distinct and permanent budget exists for supply chain human resource-strengthening activities (i.e., training, incentives, coaching, performance management) | Workforce plans are updated annually and used to inform recruiting and other staffing decisions and mentoring and coaching are based on either a performance development and/or succession plan |
| ✓ | ✓ | ✓ | Roles and responsibilities | Roles and responsibilities are not clearly defined | Job tasks are based on applicable competency model and are used for recruiting | Roles and responsibilities are clarified and documented | Sufficient material management (supply chain) staff | High-performing teams are formed and supply chain managers are empowered |
| ✓ | ✓ | ✓ | Innovation | Basic approaches to supply chain improvement and problem solving | Innovative ideas tolerated and utilised to improve supply chains as well as solve supply chain issues | Supply chain personnel have expertise and are able to improve supply chain operations as well as define and solve complex problems | Skilled staff with proactive approach to evidence-based quality improvements | Systematic thinking capabilities and Strategies to promote innovation and creativity among supply chain staff |
| ✓ | ✓ | ✓ | Incentives and working environment | Motivational mechanism and rewarding strategies to encourage enhanced performance (financial and non-financial). | Motivation to supply chain staff for proper and accurate demand and inventory data collection, reporting, and analysis | Good working environment and working conditions support performance | Workers have good work culture and have a sense of ownership for their roles and are motivated to do their jobs | Supply chain staff has high accountability, engagement and empowerment |

**Table A1.** *Cont.*

| Key Supply Chain Nodes and Capabilities Profile | | | Maturity Model Capabilities | Maturity Model Stages | | | | |
|---|---|---|---|---|---|---|---|---|
| Manufacturing | Distribution | Clinics/Hospitals | | Initial Unpredictable and ad hoc Process | Repeatable Disciplined Processes | Defined Quantified and Predictable Process | Managed Quantified and Predictable Process | Optimised Continuous Improvements |
| Human resources | | | | | | | | |
| ✓ | ✓ | ✓ | Training and education | A human resource information system is regularly updated at all applicable levels and is used to make human resource decisions | Recognised need to invest in talent development through continuous programs | Performance development plans exist for all supply chain personnel and are regularly reviewed An education and development framework and programme is in place which directs and coordinates the delivery of a learning environment and continuous training for staff | Personnel with supply chain expertise guide and inform supply chain-related strategies, and policies and staff have good supply chain planning capabilities and are able to use the available information for informed planning processes | Sufficient budget and authority given to senior-level SC champions to fully empower the development of SC staff |
| Distribution management | | | | | | | | |
| ✓ | ✓ | ✓ | Risk management | No framework detailing all potential supply chain risks and associated contingency plans | Formal risk assessment framework in place | Framework to support risk analysis, evaluation, and weight factors applied to risks | Contingency plans to mitigate supply chain risks and supply chain disruptions in place | Predictive analytics capabilities utilised to timely mitigate supply chain risk |
| ✓ | ✗ | ✗ | Demand planning | Demand plan is based on statistical forecasts, historical data and trends | Judgements and experience of demand planners included in the demand plan | Synchronised joint demand planning with other external supply chain partners utilising inventory and consumption pattern data, and the demand plan is tracked for accuracy | Demand planners have the right skill set (quantitative, computer, interpersonal, and process management) | Demand-driven planning based on actual customer consumption The demand plan is included in a supply chain platform and is continuously updated using real-time demand information |

**Table A1.** *Cont.*

| Key Supply Chain Nodes and Capabilities Profile | | | Maturity Model Capabilities | Maturity Model Stages | | | | |
|---|---|---|---|---|---|---|---|---|
| Manufacturing | Distribution | Clinics/Hospitals | | Initial Unpredictable and ad hoc Process | Repeatable Disciplined Processes | Defined Quantified and Predictable Process | Managed Quantified and Predictable Process | Optimised Continuous Improvements |
| Distribution management | | | | | | | | |
| ✓ | ✓ | ✗ | Supply planning | Supply plan created based on inventory policies | Supply plan is created from a basic demand plan and does not make use of master data | Integrated organisational supply planning created that is based on data within the whole organisation (lead-time, supplier constraints and stock-holding, consumption data) | Supply plan is included in the standardised digital supply chain platform that is accessible to relevant supply chain partners | Automatic supply order generation by the digital supply chain platform, and supply chain staff just authorise |
| ✓ | ✓ | ✗ | Distribution planning | The location (geography) of customers is documented | Delivery scheduling and route planning using a planning tool to assign transport requests | 3PL service level agreements defined and their performance is measured | Replenishment orders generated from actual end customer demand | Strategic capacity planning (warehouse, labour, and transport capacity) |
| ✓ | ✓ | ✓ | Inventory management | The inventory management system is entirely manual | Dynamic policy on how much inventory should be kept at the warehouse with defined min/max levels | Lean practices (5S) implemented and owned by all, to ensure a clean, safe, and efficient workplace in the warehouse | A simulation tool is regularly used to review layout and labour requirements aligned with demand | Segments out of stock monitoring and product out-of-stock in the segments (not OOS in the warehouse) are a key performance indicator in the warehouse metrics |
| ✓ | ✓ | ✓ | Warehouse visibility and automation | Technologies and data are leveraged to improve warehouse operations such as the ability to locate specific product within warehouse and tracking of both inventory levels and order status | Basic warehouse automation using barcodes to ensure warehouse visibility | Advanced warehouse automation and mechanisation leveraged (RFID technologies) | warehouse management system utilised and real-time exception management performed through alert messaging | Warehouse management systems integrated with the standardised supply chain platform |

**Table A1.** *Cont.*

| Key Supply Chain Nodes and Capabilities Profile | | | Maturity Model Capabilities | Maturity Model Stages | | | | |
|---|---|---|---|---|---|---|---|---|
| Manufacturing | Distribution | Clinics/Hospitals | | Initial Unpredictable and ad hoc Process | Repeatable Disciplined Processes | Defined Quantified and Predictable Process | Managed Quantified and Predictable Process | Optimised Continuous Improvements |
| Distribution management | | | | | | | | |
| ✓ | ✓ | ✓ | Order management | Formal standards for order management | Procedures in place to review, process, and prioritise customer orders and to ensure customer order accuracy | Integrating customer order management with supply chain planning and execution processes | Holistic, real-time order process support with value-added functions like manufacturing/distribution capacity reservation or order configuration based on a collaborative design and execution portal | Integrated supply chain platform exists to identify, track, and manage each customer order |
| ✓ | ✓ | ✗ | Logistics and transport flexibility | Distribution operation is customer-driven (oriented to customer service i.e., on-time and in-full delivery) | Distribution operation is well executed with dynamic dispatching based on customer orders received | Delivery variances and root causes used to define action plans/adjust plan or execution, depending on the root causevariance and root cause analysi | Transport arrives on time for warehouse/facilities appointments with defined service-level agreements that are measured and managed | Proof of delivery (POD) used and returned to the warehouse |
| ✓ | ✓ | ✓ | Expiry Management | Procedures in place for expired healthcare products management | Norms and standards for healthcare products wastage are defined | Tracking of expiry dates for medicines and wastage risk | Established picking policy for expired medicines (FEFO) | Expired medicine quantities are measured, managed, and reported on the digital supply chain platform that is accessible to relevant supply chain partners, and the policy for expiry management continuously audited |

Background color is to make the seperation between columns clear.

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
