# Peer review of "A Network Maturity Mapping Tool for Demand-Driven Supply Chain Management: A Case for the Public Healthcare Sector"

_sustainability, doi:10.3390/su132111988_

Round 1

Reviewer 1 Report

file attached

Author Response

Point 1: all equations must be numbered and cited in the text.

Response 1: Done

Point 2: all figures and tables must be cited in the text

Response 2: Done

Point 3: the bibliography is old and should be revised

Response3: updated with additional  and new references on the healthcare supply chain network (section 2.1, new section)

Point 4: this item, 2.2.Maturity models, should be expanded.

Response 4: Updated with key concepts from the papers suggested.

General, A new section added, to improve the context and position the problem clearly. A new section on gap analysis and contribution added (section 5). The conclusions enhanced and updated.

Reviewer 2 Report

The paper focuses on the use of Demand Driven Supply Chain Management operating practice in the healthcare industry. Healthcare supply chain is a complex adaptive ecosystem where low forecast accuracy and high demand volatility have resulted in an increase in stock outs, operational inefficiencies, poor health outcomes and a significant increase in supply chain costs. To cope with these challenges the authors analyze the implementation of DDSCM practices.

The topic is interesting.

I suggest the authors emphasize the issues of demand volatility which can be partially managed through Demand Driven Supply Chain Management. In fact, I found only in two parts mentioned this issue, but in my opinion, it needs to be stressed since in several sectors, and also in healthcare sector, it is a big issue (Akbarpour et al., 2020; Carbonara and Pellegrino, 2018; Nasrollahi and Razmi, 2021; Pellegrino et al., 2020).

In the conclusion sector the authors need to stress their contribution to the literature and practice.

In the abstract the authors need to explicit the acronym DDSCM where they mention for the first time Demand Driven Supply Chain Management.

Suggested references

  • Akbarpour, M., Torabi, S. A., & Ghavamifar, A. (2020). Designing an integrated pharmaceutical relief chain network under demand uncertainty. Transportation Research Part E: Logistics and Transportation Review136, 101867.
  • Carbonara, N., & Pellegrino, R. (2018). Real options approach to evaluate postponement as supply chain disruptions mitigation strategy. International Journal of Production Research56(15), 5249-5271.
  • Nasrollahi, M., & Razmi, J. (2021). A mathematical model for designing an integrated pharmaceutical supply chain with maximum expected coverage under uncertainty. Operational Research21(1), 525-552.
  • Pellegrino, R., Costantino, N., & Tauro, D. (2020). The value of flexibility in mitigating supply chain transportation risks. International Journal of Production Research, 1-18.

Author Response

Comment 1: I suggest the authors emphasize the issues of demand volatility which can be partially managed through Demand Driven Supply Chain Management. In fact, I found only in two parts mentioned this issue, but in my opinion, it needs to be stressed since in several sectors, and also in healthcare sector, it is a big issue (Akbarpour et al., 2020; Carbonara and Pellegrino, 2018; Nasrollahi and Razmi, 2021; Pellegrino et al., 2020).

Response 1: Authors included the suggested references to clearly discuss the concept of demand volatility. (New section: Section 2.1)

Comment 2: In the conclusion sector the authors need to stress their contribution to the literature and practice.

Response 2: Updated (page 37, last paragraph)

Comment 3: In the abstract the authors need to explicit the acronym DDSCM where they mention for the first time Demand Driven Supply Chain Management.

Response 3: Updated in the abstract

Reviewer 3 Report

The paper provides a network level analysis when addressing DDSCM in the healthcare industry. It presents a network-maturity mapping tool to assist unique supply chain nodes in public healthcare. The main novelty of contribution of this paper lies in the combination of three
research areas (DDSCM, Maturity models and Healthcare) thereby contributing to the body of research and the public healthcare sector. Key findings show that collaboration across all nodes in the public healthcare SCN is in its infancy.

The list references used in the article seems to be a little bit outdated. It would be useful to add more actual and relevant references to the topic. The continuity of the chapters is appropriate. The characteristics of exploratory and examined problem should be clarified more clearly. Determination of the main aim and research questions should be described more in detail, so as the process of searching appropriate methods and reasons of chosen methodology of applied methods. The aim of the paper is sufficient but does not bring any impressive results. The research questions should be argued and supported by more citations from the literature. The discussion part of the paper should be extended, it does not explain clearly how this study contributed to a theoretical and practical level of the research.

Author Response

Comment 1:The list references used in the article seems to be a little bit outdated. It would be useful to add more actual and relevant references to the topic.

Response: New relevant references added.

Comment 2: The continuity of the chapters is appropriate. The characteristics of exploratory and examined problem should be clarified more clearly. Determination of the main aim and research questions should be described more in detail, so as the process of searching appropriate methods and reasons of chosen methodology of applied methods.

Response 2: (page 2)

Main Research question

How can a healthcare supply chain network deal with low forecast accuracy and high demand volatility using DDSCM practices that have been applied in other industries?

What are the components that make up DDSCM?

What does a network maturity mapping tool for DDSCM look like?

Is the network maturity tool for DDSM Applicable in heathcare supply chain contexts

Section 2.1 added to outline context and support the research question.

Methods used include: Literature review: to identify components of DDSCM, approaches to supply chain optimisation which include maturity models and supply chain mapping. The research questions justify the appropriate methods used. The main aim of this paper is to design a network maturity mapping tool for DDSCM that is applicable in the healthcare context to support supply chain optimization efforts.

Comment 3: The aim of the paper is sufficient but does not bring any impressive results. The research questions should be argued and supported by more citations from the literature.

Response: Added section 2.1 to clearly outline the problem and added new citations as well. The results are in 3 fold in response to the research questions;

  • Validate the components of DDSCM using experts in the DDSCM Domain
  • Validate the DDSM Framework using experts in the healthcare supply chain industry
  • Application of the network maturity tool in the healthcare sector to test its applicability and get insights in the actual environment of decision makers

Comment 4: The discussion part of the paper should be extended, it does not explain clearly how this study contributed to a theoretical and practical level of the research.

Response 4: Section 5 added to discuss the gap analysis and the contribution of the paper.

The main contribution of the practical case study approach is that it outlines the validity and applicability of the maturity mapping tool in the healthcare sector and it also provides insights from the actual decision makers for healthcare supply chain management. The results show that the tool can be used to assess as is DDSCM processes as well as guide the improvement roadmap towards best practice. This study combined 3 re-search domains, healthcare, DDSCM and maturity models to create a theoretical contribution to the exiting literature on healthcare supply chain optimization and the and the di-versified field of maturity models.

Round 2

Reviewer 1 Report

no comments

Reviewer 2 Report

The authors have addressed my previous comments.